# Characteristic Analysis of a Chipless RFID Sensor Based on Multi-Parameter Sensing and an Intelligent Detection Method

**DOI:** 10.3390/s22166027

**Published:** 2022-08-12

**Authors:** Luyi Liu, Lan Chen

**Affiliations:** Department of Electrical and Electronic Engineering, Shanghai Institute of Technology, Shanghai 201418, China

**Keywords:** chipless radio frequency identification, multi-parameter sensor, structural health monitoring, temperature and humidity sensing, wireless detection system

## Abstract

Chipless radio frequency identification (RFID) technology has been widely used in the field of structural health monitoring (SHM), but most of the current research mainly focuses on the detection of mechanical properties and there are few studies on the multi-physical parameters (for example, temperature and humidity) in the climatic environment around the structure. Thus, it is necessary to design a small and compact sensor for multi-parameter detection. This paper proposes a multi-parameter chipless RFID sensor based on microstrip coupling, which supports 4-bit ID code and integrates two detection functions of temperature and humidity. Through linear normalization fitting, the sensitivity of the sensor is about 2.18 MHz/RH in the ambient relative humidity test and the sensitivity of the sensor is about 898.63 KHz/°C in the experimental test of water bath heating from 24.6 °C to 75 °C. In addition, this paper proposes an engineering application detection method, designs a lightweight dynamic spectrum detection and wireless transmission platform based on a lightweight vector network analyzer (VNA) and realizes the real-time extraction and transmission of RFID spectrum sensing data. The means are more flexible and economical than traditional experimental scenarios.

## 1. Introduction

In recent years, the integration of chipless radio frequency identification (RFID) technology in the Internet of Things (IoT) has facilitated the structural health monitoring of modern industrial systems’ critical infrastructure [1]. Structural health monitoring (SHM) is a combination of sensor technology and the Internet of Things (IoT) to realize an automatic detection system for structural damage to civil infrastructure. However, the majority of the current research on structural health monitoring is still limited to stress characteristics [2,3,4], and the monitoring of physical parameters in the environment where the structure is located, such as temperature and humidity, gas content, pH and other parameters, has not yet been fully researched. Therefore, it is necessary to develop a single chipless RFID sensor for multi-physical parameter sensing.

RFID technology utilizes wireless signals to identify the target. Currently, chipless frequency domain RFID uses resonant structures to encode data into the frequency spectrum and is associated with ID via basic coding elements corresponding to pre-designed simple shapes, and the functions of signal reception, signal processing and signal transmission are no longer separated in geometry and concept, but coupled within the same structure. Consequently, the cost, dependability and recyclability of tags have increased. With the development of the IoT era, RFID technology has opened a new paradigm, especially the combination of information sensing and the IoT, which has greatly stimulated the potential in the field of structural health monitoring [5,6].

Combining novel sensing materials and chipless RFID sensors to create multi-parameter chipless RFID sensors has become the subject of research [7] with the advent of high-performance composite materials. The interaction of environmental physical elements with chipless RFID sensors alters the dielectric constant of the sensitive material, resulting in a shift in the resonant frequency and a corresponding change in the amplitude of each sensor element [8,9]. Typically, researchers combine sensing materials into chipless RFID tags using two distinct ways. The first strategy is to employ sensing materials as substrates. When physical parameters associated with the surrounding environment change, the substrate’s dielectric constant also alters. Additionally, implement sensing and the detection of the surrounding environment by the modification of involved parameters such as the sensor’s resonance frequency. M. Borgese et al. proposed a unique inkjet-printed chipless RFID humidity sensor in 2017, which was inkjet-printed on a thin sheet of Mitsubishi paper and passed through the Mitsubishi paper’s dielectric. The change in the constant causes the frequency of the resonance peak to shift, allowing humidity to be monitored [10]. In 2021, Nimra Javed et al. fabricated a 12-bit tag with a humidity-sensitive Kapton HN substrate by deploying a multi-walled carbon in the center of the tag nanotubes (MWCNTs) to change their electrical conductivity through CO2 gas sensing properties, allowing humidity and CO2 gas to be monitored simultaneously [11]. The second method is to form a polymer substance with sensitive qualities into a film shape and cover the surface of an existing chipless RFID tag in order for it to function. Changes in the physical components of the surroundings cause changes in film-related parameters, which are subsequently transferred to the sensor’s corresponding resonance. In terms of frequency, the sensor’s frequency shift characteristic is employed to define changes in physical parameters in the surrounding environment [12,13]. Polyvinyl alcohol (PVA) is a water-soluble polymer with numerous applications. It is easily soluble in water because it forms hydrogen bonds quickly with water. P.Bergo and others discussed the PVA film production technique and the relationship between PVA dielectric constant and water content [14]. Because of its temperature-sensitive characteristics, reduced graphene oxide (rGO) is commonly employed to construct temperature-sensitive sensors. Using rGO as a substrate or combining rGO materials with other materials to inspire its potential in the field of temperature sensing [15].

With the evolution of sensors and micromachining technologies, it has become possible to integrate multi-parameter sensors. This sensor may simultaneously measure multiple parameters, and each sensor element is independent of the others. Adding the coding function enables the effective identification of a single sensor when numerous sensors are present. Therefore, numerous researchers have explored the multifunctional integration of multi-parameter sensing and tag identification. Emran M. Amin created a multi-parameter chipless RFID sensor with integrated coding, humidity and temperature monitoring in 2016 by combining temperature-sensitive Phenanthrene and humidity-sensitive polyvinyl alcohol. As an ID encoder and an electric field LC-coupled (ELC) resonator coated with a smart material layer, temperature and humidity sensing and information encoding are realized in a small planar configuration [16]. The complementary split ring resonator (CSRR) structure was initially introduced by Pendry et al. and was subsequently widely utilized to characterize changes in parameters such as stress, temperature, humidity and gas [17,18,19]. Ma et al. created a chipless sensor based on frequency domain coding by incorporating CSRR resonant units of varying sizes on a microstrip line, which greatly improved frequency band utilization [20].

The employment of intelligent materials in conjunction with chipless RFID sensors has significantly increased the potential of tag sensors in application scenarios. Presently, the majority of research on the sensitive qualities of ambient physical factors focuses on the construction of RCS chipless RFID tag sensors, among which ELC-type resonators are extensively employed due to their capacitance-oriented design. The chipless RFID tag based on the microstrip transmission line has a bigger data capacity than the RCS tag based on backscattering, the mutual coupling interference can be alleviated by adjusting the distance between different resonant units and the cross-polarized transceiver antenna can be matched by retransmission. Without the need for cumbersome horn antennae, detection is conducted [21,22].

In this paper, a multi-resonator cascade design based on a 50 Ω microstrip transmission line is implemented, and a coupling sensor tag based on CSRR and ELC resonator is proposed. Four different-sized CSRR resonant units are intended for the identification of tag identity information. An additional CSRR resonator is installed to characterize the environmental temperature sensing, at the same time, an ELC resonator is integrated to improve the humidity sensing capability of the tag. Then, the variation law of the amplitude and shift of the resonant frequency peak of the multi-parameter sensor under different temperature and humidity conditions was discussed. The contributions of this paper are summarized as follows. (I) A chipless RFID multi-parameter sensor is designed, which realizes the integration of three functions of temperature, humidity and encoding. The three functions of the designed sensor are independent of each other and can be combined flexibly. (II) Through the linear normalization fitting of the resonant frequency shift characteristics of the tag, its good temperature and humidity sensing characteristics are verified. (III) An engineering-focused, lightweight technique for wireless dynamic spectrum detection is proposed.

## 2. Design of Chipless Sensor Based on Microstrip Coupling Resonance

### 2.1. Theory of Designing Microstrip Resonators

Microstrip patch antenna is a good choice for sensor fabrication because of its simplicity and low production cost. The main characteristic of the microstrip antenna is that it has more physical parameters than conventional antenna, so it can be designed in various shapes and sizes. Based on the frequency domain characteristics of microstrip antenna, the resonant frequencies of multiple parameters are realized in the working frequency band by designing a coupling cascade of multiple resonators and microstrip transmission lines. The resonant frequency of the microstrip resonator is closely related to the electrical length of the designed resonator, and the relationship between them can be given by Equation (1) [23]:(1)$$fres=\frac{1}{\sqrt{\varepsilon_{eff}}}\frac{nc}{2L_{res}}$$
where *c* is the speed of light, *ε_eff_* is the effective permittivity, *L_res_* is the effective electrical length of the resonator and *n* is the *n*th resonant mode.

### 2.2. Design of CSRR and ELC Resonators Based on Microstrip Coupling

CSRR symmetrical structure resonator is a resonant structure with high-quality factor (Q value) and generates a very compact frequency bandwidth, so it can efficiently implement more resonant unit settings in the same frequency band [24].

In the frequency domain, a kind of composed of multiple complementary split ring resonators is presented in this paper, which the CSRR resonance unit with the main coupled microstrip line feeds. Its structural layout is shown in Figure 1a. In order to better compare the sensitivity of the proposed CSRR resonant unit to ambient temperature and humidity, an existing reported ELC resonator with good capacitance characteristics is combined in Figure 1b. As shown in Figure 1b, a CSRR resonant unit and an ELC resonant unit are simultaneously set on the main microstrip transmission line in the tag. Figure 1 shows the main parameter dimensions of the tag design. In this work, Rogers 5880 is used as the dielectric substrate and the optimization design is carried out in the radiofrequency (RF) simulation software HFSS. Table 1 shows the specific values of each parameter after tag optimization.

Figure 2 shows the equivalent circuit of the CSRR and ELC resonators coupled simultaneously on the microstrip line. The *L*11 represents the equivalent inductance on the main transmission line. The greater the value of *L*11, the greater the loss of signal, which means that the narrower the microstrip line, the greater the attenuation of the signal. *C*11 and *C*12 represent the degree of coupling between the microstrip line and the CSRR and ELC resonators, respectively. The smaller the gap between the resonator and the microstrip line, the greater the degree of coupling, and the greater the return loss obtained at the resonant frequency. The bandstop resonance characteristics of the resonator can be modeled with lumped-circuit elements, where the *L*21, *C*21 equivalent elements are used to generate the resonant frequency of the CSRR resonator, and the *L*31, *C*31 equivalent elements are used to generate the resonant frequency of the ELC resonator. The simplified circuit model of the ELC unit resonant frequency and the CSRR unit resonant frequency is $\omega_{r}=\sqrt{1/(L_{r}C_{r})}$. Where the equivalent inductance and capacitance of CSRR can be derived as follows [25,26,27]:(2)$$C_{0}=\varepsilon_{0}\frac{K\sqrt{1-k^{2}}}{K(k)}$$
(3)$$k=\frac{d/2}{w+d/2}$$

Here, *K* (*k*) is the first complete elliptic integral, *d* is space of the CSRR inner and outer rings, *w* is the line width of the CSRR microstrip frame and *C*_0_ is the capacitance of unit length of the two split ring.
(4)$$C_{21}=C_{0}[L+3(d+w)/2]$$
(5)$$L_{21}=\frac{\mu_{0}}{2}\frac{l_{avg}}{4}4.86\left[\ln\left(\frac{0.98}{\rho }\right)+1.84\rho \right]$$

Attention coefficients $\rho =\frac{w+d}{L-w-d}$ is the filling factor, $l_{avg}=8[\frac{L}{2}-\frac{\left(w+d\right)}{2}]$.

Figure 3a shows the scattering characteristic curve of a single CSRR and the coupling structure when the distance dg1 of the microstrip transmission line is 0.1 mm. When the length L1 of CSRR is 9.6 mm, one of the better bandstop resonance curves is formed at 2.95 GHz, the transmission coefficient S21 is about −21 dB and the reflection coefficient S11 is about 0 dB. Figure 3b shows the S11 and S21 parameter response curves of a single CSRR and a single ELC coupled with the main microstrip transmission line at the same time, where the resonant frequency generated by CSRR is about 3.26 GHz, and that generated by ELC is about 3.95 GHz. The resonant bandwidth of CSRR is about 5 MHz, which is less than the 9 MHz bandwidth of ELC, so the notch curve is more compact. Moreover, CSRR has lower resonant frequency and higher return loss than the ELC resonator when the size difference between the two is not much.

Figure 4 compares the surface electric field distribution of the ELC resonator and CSRR resonator at operating frequencies. Figure 4a shows the surface voltage distribution at a single CSRR operating frequency. In the CSRR structure, due to the setup of the multiple split ring and each division characteristic of the microstrip ring stopping the current through the loop, both sides inside the CSRR resonant unit can create two equal capacitors. Figure 4b shows the surface electric field distribution at their respective operating frequencies when the CSRR resonant element and ELC resonant element are coupled with a microstrip line at the same time. It can be seen from the electric field distribution that the electric field of the ELC is mainly concentrated in the capacitor plate accessories, while the electric field of CSRR is mainly concentrated among multiple capacitor plates inside the resonator.

Smart polymer materials are very sensitive to physical parameters in the environment; thus, selecting appropriate smart polymer materials combined with the frequency-shifting properties of the chipless RFID sensors can be used to analyze the behavior of the physical parameter changes in various environments. Figure 5 simulates the shift in the resonant frequency of the sensor that may be caused by changes in temperature and humidity in the environment when the resonator surface is coated with a sensitive polymer material, the analysis shows that the proposed ELC resonator has more obvious advantages over the sensitive shift characteristic of the CSRR resonator, but the size of the microstrip line of the ELC resonator is smaller and the resulting resonance frequency is higher. However, the CSRR resonator can achieve a larger size microstrip line in the same physical area than the ELC resonator, so it can generate a lower resonant frequency, and because the CSRR resonator has higher Q values and narrower impedance bandwidth, so more resonant frequencies can be set on the same frequency band, which effectively improves the utilization rate of the corresponding working frequency band. Therefore, combined with the respective advantages of the ELC and CSRR resonators, this paper designs a hybrid resonator RFID tag based on microstrip coupling, which can realize multiple sensing functions in a compact bandwidth.

### 2.3. 6-Bit Chipless RFID Tag Design

CSRR structure is a common structure used to characterize the characteristics of parameter sensing. However, the research on the application of the CSRR resonator based on microstrip coupling in temperature and humidity sensing is not sufficient at present, and the analysis of its sensitive characteristics and the design of multi-functional integration are very few. Therefore, this paper proposes a 6-bit chipless RFID multi-parameter sensor. The sensor is based on the CSRR resonator and microstrip line coupling layout. To achieve multiple functions in a compact bandwidth, multiple CSRR resonators are set in the tag, and the existing ELC resonator is set as a reference and comparison. As shown in Figure 6, the tag structure and main parameter dimensions are designed in this paper, in which the metal thin layer is used as the radiation sheet and the metal layer on the surface is attached to the dielectric substrate. In addition, Rogers 5880 (relative permittivity ε_r_ = 2.2, h = 0.79 mm, tan δ = 0.0009), which has a low dielectric constant, was used as the substrate in the design of the resonator to improve the effect of the sensitive coating.

Figure 7 shows the S21 resonance response curve obtained by optimizing the design of a multi-parameter sensor in the RF simulation software HFSS, and the main parameter dimensions of the optimized tag structure are provided in Table 2. The sensor generates a total of six resonant frequencies in a small frequency band, among which five lower resonant frequencies are generated by the CSRR resonant unit, which are 3.03 GHz, 3.14 GHz, 3.28 GHz, 3.41 GHz and 3.61 GHz, respectively, and the ELC resonant unit is set at the highest 3.96 GHz. The working bandwidth of more than 110 MHz is reserved between each resonant unit. The CSRR resonator with the lowest resonance frequency placed in the reverse position is selected as the temperature and humidity sensor, and the remaining four CSRR resonant units are used for the information coding of the tag. Each CSRR bandstop resonator corresponds to a resonant frequency in the spectrum, and the resonant frequency corresponds to a logical “1” state of the encoding bit. When there is no bandstop filtering in the corresponding frequency band, the encoding bit logic turns to “0” state, so the tag can generate 4-bit binary coding and 16 kinds of identity information. Figure 8 shows the actual image of the proposed tag antenna fabricated on the Rogers 5880 dielectric substrate using thermal transfer technology under laboratory conditions, and the tag S21 response curve measured by a vector network analyzer (SYSJOINT SV4401A) is shown in Figure 8.

## 3. Experimental Testing and Analysis of Sensor Behavior

### 3.1. Relative Humidity Experiment

The selection of humidity-sensitive materials as substrates can reflect the changes in relative humidity in the environment to a certain extent; however, at present, few pressure-resistant materials in the field of structural health monitoring (SHM) have good sensitivity to changes in relative humidity at the same time. In this experiment, the aim is to explore a multi-parameter sensor design that is easy to integrate and has good compression performance.

It is a feasible solution to monitor the relative humidity in the environment by affecting the frequency shift of the resonator by attaching the PVA coating of the humidity-sensitive material on the sensor surface. Moisture is absorbed by the material coating, and then mapped to the corresponding resonant frequency of the sensor. With the change in the relative humidity in the environment, the relative permittivity of the PVA coating will also change, thereby affecting the peak value of the sensor’s resonant frequency, resulting in certain frequency shift effects.

In order to analyze the humidity sensitivity of the tag, this work used the PVA colloidal film and transferred it to the resonance unit of the sensor with a small amount of adhesive, and simulated the change in environmental relative humidity in the closed climate box as shown in Figure 9a. The tag is connected to the vector network analyzer (VNA) through two low-loss RF cables running through the box, and an adjustable humidifier is placed in the airtight box. In order to ensure the uniform change in humidity, a plate of anhydrous CaCl_2_ hygroscopic agent is also installed in the box to balance the rising trend of humidity, and a high-precision probe-type electronic temperature and humidity instrument is used to monitor the change in humidity in the box in real time. Figure 9b shows the actual humidity test scenario, the gap of the box was completely closed with tape (3M) and tissues, the initial temperature is 24.6 °C and the initial humidity is 27%. Considering that the relative humidity in the actual environment generally ranges from 30% to 90%, the relative humidity test starts from 30%, and the prepared PVA film is pasted on the ELC and CSRR resonant units, respectively.

Figure 10 shows the frequency shift of the ELC resonant unit with the increasing relative humidity in the environment. The resonant frequency corresponding to the ELC resonant unit without PVA film on the surface is 3.96 GHz. When the PVA gel film is covered, the initial frequency will have a certain deviation. Since it is hard to measure the moisture on the surface of the PVA film, it is first air-dried for 2 days and then placed in a sealed box with a relative humidity of about 30% for 2 days to perform a certain dehydration treatment [28].

The shifts of all resonant frequencies can be observed as a whole from Figure 10a, and the shifts of the ELC resonant units are partially enlarged in Figure 10b, from the initial 3.8814 GHz to 3.7714 GHz, the resonant frequency is reduced by about 110 MHz, and the change in the resonance curve is more obvious in the high humidity range where the relative humidity is greater than 70%. As shown in Figure 10c, the humidity sensitivity of the ELC resonator is about 2.18 MHz/RH by linearly normalizing the humidity sensing characteristics of the ELC resonator, and the experiments show the best linearity in the range of 30–60%RH, so the optimum resonant frequency range corresponding to the humidity sensitivity is about 3.817–3.881 GHz.

Figure 11 shows the frequency shift of the CSRR resonant unit with the increasing relative humidity in the environment. After placing PVA gel film on the surface of the CSRR resonator, it is also put into the sealed box and put in a static position for two days. From Figure 11a, we can observe the overall shift of all resonant frequencies. In Figure 11b, the resonance frequency shift effect is partially amplified and the selected frequency band is quantified as the humidity sensing region, which is shifted from the initial 2.69 GHz to the 2.615 GHz and the frequency reduced by 75 MHz. When the humidity increases, the increase in water accumulation on the surface of the tag will have a certain impact on the coupled resonant unit. It can be seen from the analysis that the resonant frequency shift is relatively uniform with the increase in humidity, but the resonance effect of the bandstop characteristic in the high humidity area is obviously reduced. When the PVA film absorbs enough water, the penetration depth and shift effect of the resonant frequency will increase, and the notch depth will become significantly shallower under high humidity, which is no longer easy to capture. The reason is that as the relative humidity of the environment increases, the relative permittivity of the coating changes and the transfer efficiency increases. Secondly, the thickness of the coating will increase to a certain extent when it absorbs water in a humid environment, which is also related to the thickness of the film preparation.

It is worth noting that although the measurement results of the experiment are affected by multiple factors in the environment, the overall change trend is consistent [29]. According to the trend of the actual measurement result, with the increase in the ambient humidity, the absorption efficiency of the coating on the water molecules in the air increases in the high humidity area. This results in a faster increase in coating thickness. As shown in Figure 11c, the humidity sensitivity of the CSRR resonator is about 1.43 MHz/RH by linearly normalizing the humidity sensing characteristics of the CSRR resonator unit. The CSRR resonator is less sensitive to humidity than the ELC resonator, so adding a high-humidity-sensitive ELC resonator unit is a feasible solution to improve the sensor performance.

The feature of this chipless RFID humidity sensor is that it is reversible. When the PVA coating moisture is air-dried, the tag can be basically restored to the initial state and the use of a drying oven can quickly shorten the recovery time of the tag. Experiments have shown that the electrical properties of the tag are temporary with humidity changes and the tag will not be permanently damaged and can be reused. It should be noted that the response time of the sensor is affected by the thickness of the PVA film. The increase in the thickness will slow down the response time of the tag. At the same time, the expansion of the PVA polymer matrix in the high humidity range will also have a certain impact on the response time of the tag.

### 3.2. Temperature Experiment

The thermal expansion coefficient of the pressure-resistant substrate of the chipless RFID sensor currently used in the stress field is not high, so sensitivity to temperature in the environment is low. Once the temperature threshold is exceeded, it will cause irreversible changes in the electrical characteristics of the tag sensor.

Considering the above factors, this experiment adopts the method of spreading the temperature-sensitive polymer film on the surface of the microstrip resonator unit to carry out the experiment and conducts the experimental test through the process shown in Figure 12, where Figure 12a is the 3D simulation of the temperature-sensitive experimental test of the tag modeling of the experimental scene and Figure 12b shows the actual tag temperature-sensitive experimental test process. In this experiment, the water bath heating method is used to replace the traditional high-temperature climate box. Compared with the high-temperature climate box, the heat conduction method of the water bath heating method is more uniform and the temperature is controllable.

In order to more accurately simulate the temperature changes in the actual environment, this experiment places an intrusive heater in the high-temperature-resistant water tank, and the heating temperature of the heater’s water bath can be adjusted intelligently. Wrap a layer of aluminum foil with good thermal conductivity on the surface of the water tank, fix the tag on the aluminum foil with high-temperature-resistant tape (3M company, Saint Paul, MN, United States) and make the surface fully fit. The thickness of the aluminum foil is 9 μm and the two ports of the tag are wired to measure through an RF cable to reduce external interference.

In order to reduce the influence of heat loss during heat transfer, a patch temperature sensor is installed on the tag surface and an industry standard water temperature tester is inserted into the water tank to record the change in tag temperature in real time. The temperature experimental measurement can reach up to about 80 °C, but the temperature is higher than 75 °C and the region is unstable, so the measurement range is selected between 25 °C and 75 °C.

In this paper, the purchased rGO solution prepared by the thermal reduction method and PVA solution were mixed in a ratio of 1:5 and placed in a glass container and evaporated naturally at room temperature to obtain a layer of rGO and PVA composite film, which has good temperature sensitivity. In the range of 46.85 °C to 126.85 °C, the dielectric constant increases with the increase in temperature [30]. The free space method is one of the effective analytical methods for measuring the dielectric constant. The loss tangent of the composite film with temperature changes can be expressed by the following Equation (6):(6)$$\left\{\begin{array}{c} \varepsilon =\varepsilon^{\prime }-j\varepsilon^{\prime \prime }\\ \tan\delta =\frac{\varepsilon^{\prime \prime }}{\varepsilon^{\prime }} \end{array}\right.$$
where *ε* is relative to complex permittivity, *ε′* is relative to real permittivity, *ε″* is relative to virtual permittivity and tan *δ* is the loss tangent.

Figure 13 shows the relationship between the frequency response of the CSRR temperature sensing resonance unit S21 and the temperature change when the water bath is heated. The composite film of rGO and PVA was attached to the surface gap of the CSRR resonator by a small amount of adhesive, the tag was completely attached to the thermally conductive aluminum foil and both ends of the tag were connected to a vector network analyzer (SYSJOINT SV4401A). The water bath temperature was increased from room temperature 24.6 °C to 65 °C in 10 °C increments and from 65 °C to 75 °C in 5 °C increments, and the water bath temperature was held at each temperature for approximately 1–2 h with each temperature increment, a certain settling time is reserved for the tag on the aluminum foil to reach equilibrium with the temperature of the water bath, and the VNA records the data. From Figure 13a, we can observe the shift of all the resonant frequencies of the tag under the temperature change. In Figure 13b, the frequency shift effect of CSRR is partially magnified. The resonance frequency of S21 is about 3.03 GHz before the polymer film is not covered, and 3.017 GHz at room temperature after covering, which is reduced by 23 MHz. As the temperature increases, the resonant frequency shifts from the initial 3.017 GHz to 2.97 GHz, and the frequency shift reaches 47 MHz. The analysis shows that with the increase in temperature, the resonant frequency gradually decreases, and the return loss of the temperature sensor S21 also decreases continuously. As shown in Figure 14, through the linear normalization fitting of the sensing characteristics of the CSRR resonator unit in the temperature range of 24.6 °C to 75 °C, Figure 14a shows that the sensitivity of the resonant frequency of the CSRR resonator with temperature change is about 898.63 KHz/°C and Figure 14b shows that the sensitivity of the resonant frequency amplitude of the CSRR resonator with temperature change is about 0.10273 dB/°C. When the CSRR resonant unit is used for temperature sensing, the frequency shift characteristics of the resonance and the change in the depth of peak resonance have a good linearity in the measured range. Therefore, the change in temperature can be reflected from two dimensions, and the experiments show that it has the best linearity in the range of 30–45 °C, so the optimum resonant frequency range corresponding to the temperature sensitivity is about 3.003–3.016 GHz.

For a more intuitive comparison, this paper summarizes some reported research works of chipless RFID multi-parameter sensors in recent years, as shown in Table 3. This tag has the advantage of sensing multiple parameters at the same time. By combining the respective advantages of CSRR and ELC resonators, it realizes the partition setting of different sensing function working frequency bands, improves the utilization of spectrum and has good frequency shift characteristics in the measurement range, which can better monitor the changes in temperature and humidity in the environment.

### 3.3. Real-Time Monitoring and Wireless Transmission of Sensing Data

In this paper, a new method of tag spectrum detection and data extraction is proposed and the feasibility of implementing chipless RFID reader mobile detection on fixed sensor tags in the future is proposed from the perspective of engineering application. Figure 15 shows a lightweight dynamic spectrum detection and data wireless transmission architecture proposed in this paper, which mainly includes a lightweight low-cost VNA, microcontroller unit (MCU) with universal serial bus on-the-go interface (USB OTG) function, Bluetooth receiver and transmitter and passive RFID tag. In the application scenario for this experiment, the tag spectrum detection acquisition method rather than the traditional detection method is more flexible and convenient. Through the MCU controller intelligent interaction with the vector network analyzer, the Bluetooth module can support the smooth transfer data and instructions and expand the scope of the sensor, which has some potential engineering application prospects. This system is based on the Keil uVision5 software development platform and uses C language to compile programs. Using C language as a program development language, it can complete most of the functions of assembly language, with high compatibility and portability.

In Figure 16, a simple dynamic tag spectrum detection experiment scenario is set up, in which passive RFID tags are connected to VNA through RF cable, MCU communicates with low-cost VNA as the master controller and tag-scattering parameters in VNA are collected by personal computer (PC) instructions. The interaction between the MCU and the small VNA can be realized through the USB protocol. In this work, the USB OTG interface is used to connect the master and slave devices. The USB OTG interface standard allows a device to act as both a host and a slave, thus enabling bidirectional data transfer between devices. When the MCU is used as the main control device, it can realize the real-time collection of the communication device class (CDC) data of the VNA in the form of a virtual serial port and can realize the data reading of the digital electronic sensor through the inter-integrated circuit (I^2^C) bus. The scattering parameters are transmitted to PC in the plural form by MCU onboard Bluetooth communication module (HC-05). At the same time, MCU has multiple input/output (IO) interfaces and can be connected to a variety of digital sensors. The MCU in Figure 16 is connected with a highly sensitive temperature and humidity electronic sensor (DHT11) probe. The electronic sensor probe can be deployed in an airtight box to monitor temperature and humidity changes in the box in real time, which is convenient for comparison with tag test data. All data can be sent to PC for data processing and dynamic display through the Bluetooth module, and the users can select the different devices to collect the data by issuing instructions from the PC. In addition, by setting the clock module, the data can be collected regularly, which reduces the power consumption of the detection devices.

The amplitude *A* (in dB) and phase (in degrees) measured by a vector network analyzer at each resonant frequency can be expressed using the scattering parameter *S* in Equation (7) [35]:(7)$$S=10^{A/20}e^{i\varphi \pi /180^{{}^{\circ}}}$$

Therefore, the MCU controller can extract the scattering parameters in the RFID tag and obtain the amplitude information of the S parameter by collecting the complex form of the S parameter. The S21 response parameter of the two-port tag antenna sensor can be extracted using Equation (8).
(8)$$\left\{\begin{array}{c} S_{\left(2,1\right)}=\sqrt{{\mathrm{re}}^{2}S_{\left(2,1\right)}+im^{2}S_{\left(2,1\right)}}\\ S_{\left(2,1\right)}(dB)=20\mathrm{lg}S_{\left(2,1\right)} \end{array}\right.$$

As shown in Figure 16, there is still room for optimization in the passive RFID spectrum dynamic detection and wireless transmission system. By matching appropriate transceiver antennas at both ends of the two-port microstrip antenna, the wireless layout of tags and detection equipment can be realized based on retransmission, and the measurement distance of wireless detection based on retransmission in this work is about 3 cm. The lightweight chipless RFID tag detection equipment is integrated and placed on automated instruments such as drones and robots, which can realize a mobile chipless RFID detection system and can realize real-time data collection and intelligence in combination with software such as databases on the PC side processing to form a complete intelligent networked solution that can be practically applied.

## 4. Conclusions

In this article, a 6-bit chipless RFID temperature and humidity sensor based on a frequency domain is designed. The proposed CSRR resonator is arranged in the form of coupling with the microstrip line, and an ELC resonator is added as the humidity sensing unit. In this work, the humidity sensitivity of the ELC resonator and the CSRR resonator is compared and analyzed, and the temperature sensitivity of the CSRR is analyzed. The sensitivity of each sensing unit used in the tag was calculated by linear normalization fitting. The three sensing functions of the proposed RFID multi-parameter sensor are all quantified in their respective operating frequency bands, and there will be no interference between the sensing resonant frequencies of different partitions, so a flexible small-range frequency band detection can be used, which helps improve detection capability. In addition, this paper proposes a lightweight RFID tag spectrum dynamic detection and data wireless transmission architecture, which is more convenient and flexible and conducive to the development of experiments and intelligent data collection. The proposed concept provides a feasible idea to realize the possibility of chipless RFID tag detection equipment in mobile detection.

The limitation of this paper is that the experiments are carried out under a single variable condition and no further studies have been carried out on the simultaneous changes in temperature and humidity. In this work, the proposed low-cost mobile detection scheme is based on lightweight VNA, and further research needs to be carried out with higher gain transceiver antennas. In the future, customized chipless RFID readers and communication protocols will be developed and this work can greatly encourage the large-scale deployment of chipless RFID tag sensors.

## Figures and Tables

**Figure 1 sensors-22-06027-f001:**
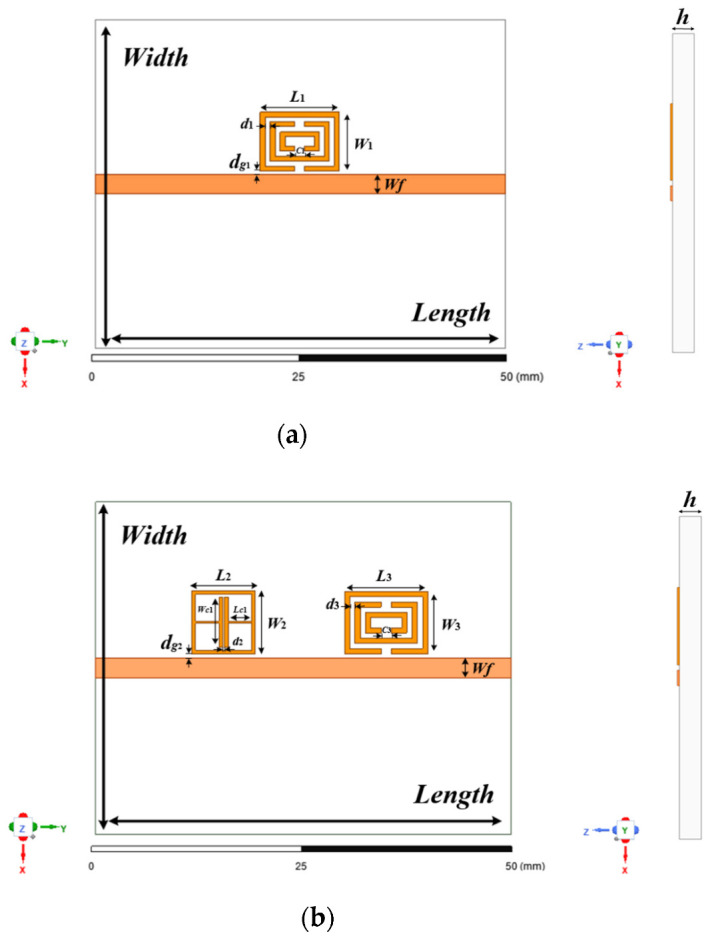
Chipless RFID tag design: (**a**) Microstrip-coupled CSRR resonator design; (**b**) The 2-bit RFID tag design.

**Figure 2 sensors-22-06027-f002:**
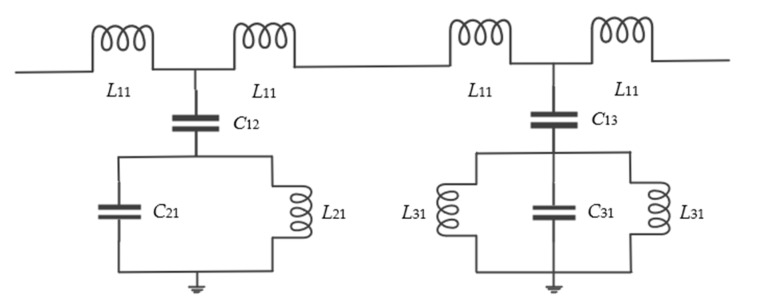
Equivalent circuit model of CSRR and ELC resonant unit.

**Figure 3 sensors-22-06027-f003:**
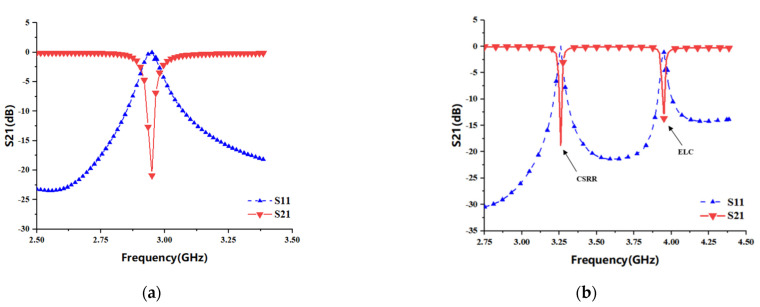
Chipless RFID tag resonance response curve: (**a**) CSRR resonance unit S11 and S21 response curve; (**b**) ELC and CSRR resonance unit S11 and S21 response curve.

**Figure 4 sensors-22-06027-f004:**
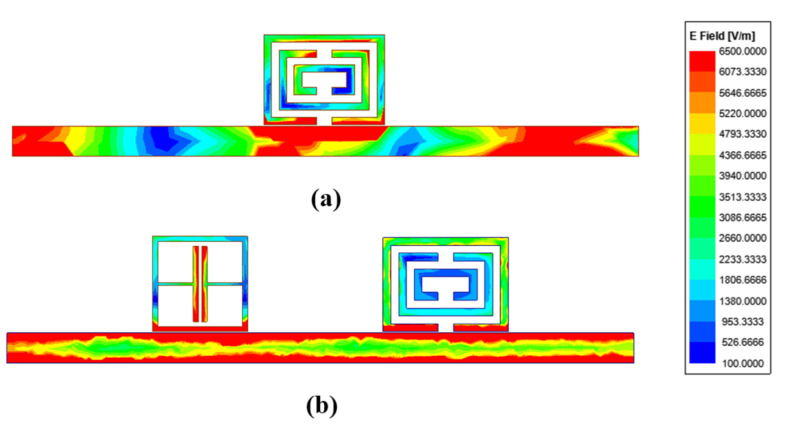
Electric field distribution of chipless RFID tags at resonance peak frequencies: (**a**) proposed CSRR resonator (2.95 GHz); (**b**) ELC resonator (3.95 GHz) and proposed CSRR resonator (3.26 GHz).

**Figure 5 sensors-22-06027-f005:**
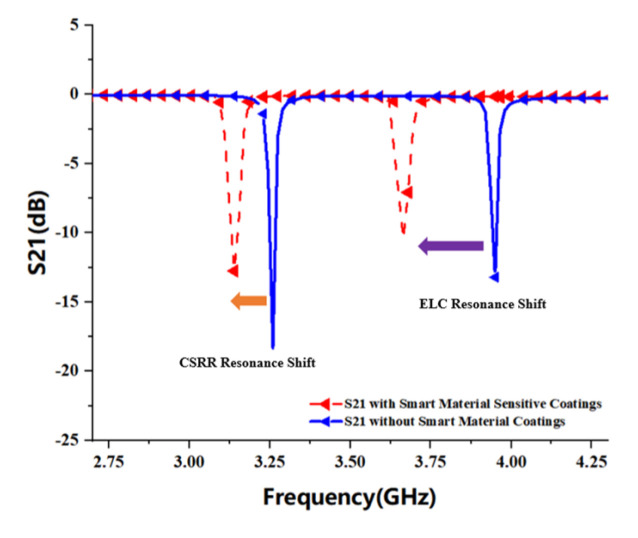
Simulate the resonance shift effect of covering polymer films.

**Figure 6 sensors-22-06027-f006:**
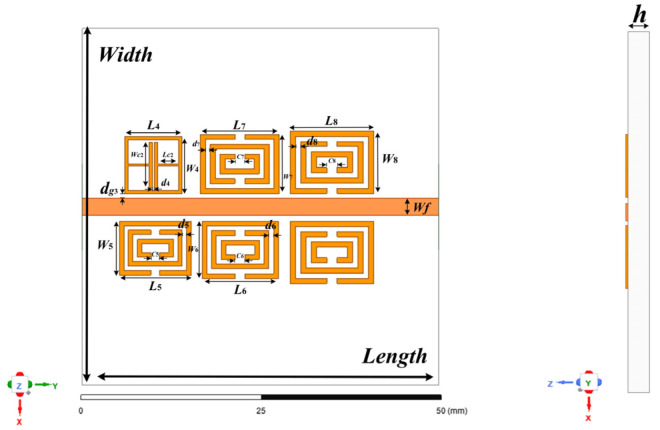
The 6-bit chipless RFID tag design.

**Figure 7 sensors-22-06027-f007:**
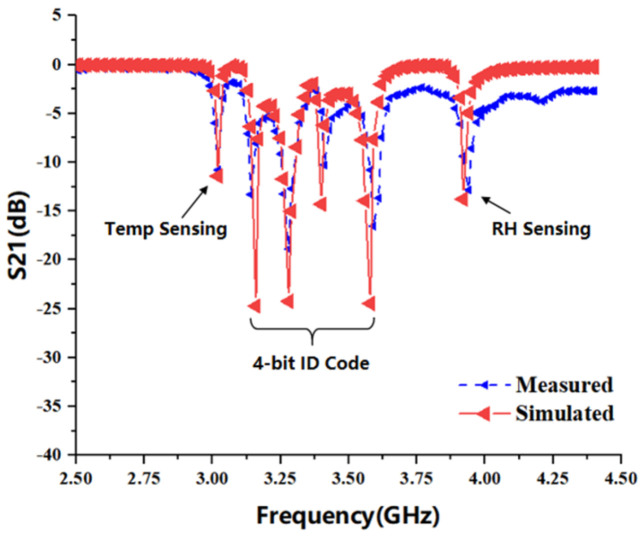
The 6-bit chipless RFID tag simulation and measurement of S21 response curve.

**Figure 8 sensors-22-06027-f008:**
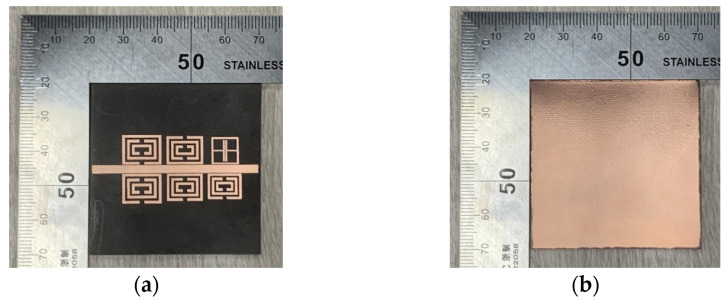
The 6-bit chipless tag object: (**a**) Tag radiation surface; (**b**) Tag ground surface.

**Figure 9 sensors-22-06027-f009:**
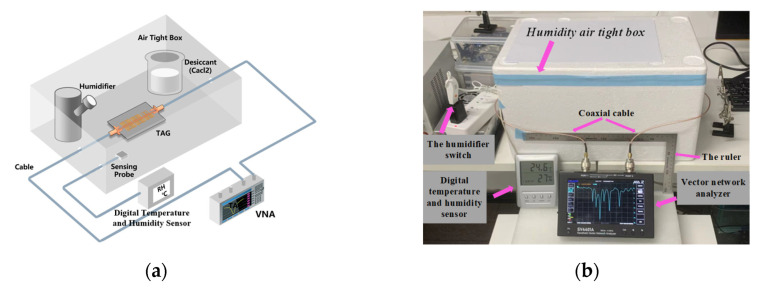
Tag humidity test scenarios: (**a**) Three-dimensional modeling experiment scene simulation; (**b**) Experimental testing of humidity in a laboratory environment.

**Figure 10 sensors-22-06027-f010:**
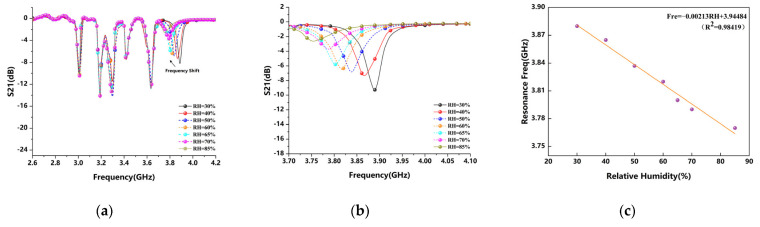
Resonant response of ELC humidity sensor unit: (**a**) Changes in the overall S21 resonant response curve of the tag sensor; (**b**) Change in the S21 curve of locally amplified ELC resonant element; (**c**) Linear fitting line of relative humidity of ELC resonant element.

**Figure 11 sensors-22-06027-f011:**
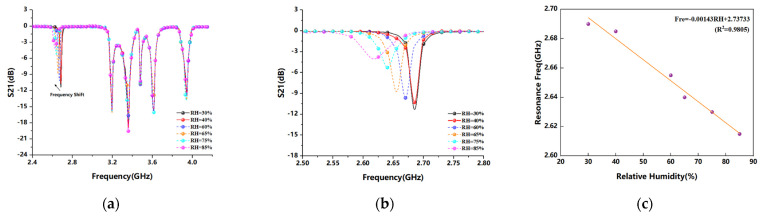
Resonant response of CSRR humidity sensor unit: (**a**) Changes in the overall S21 resonant response curve of the tag sensor; (**b**) Changes in the S21 curve of locally amplified CSRR resonant element; (**c**) Linear fitting line of relative humidity of CSRR resonant element.

**Figure 12 sensors-22-06027-f012:**
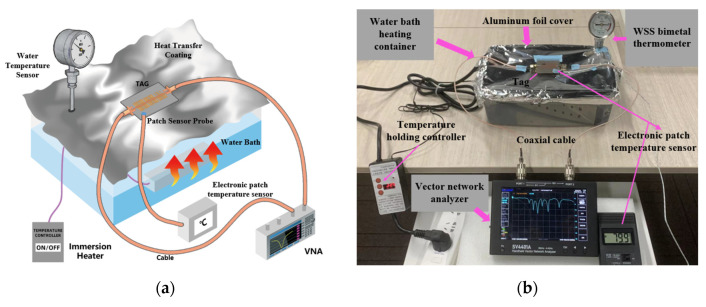
Tag temperature sensing characteristic test: (**a**) Tag temperature sensitivity test, 3D simulation experiment scene modeling; (**b**) Tag temperature sensitivity characteristic test in the laboratory.

**Figure 13 sensors-22-06027-f013:**
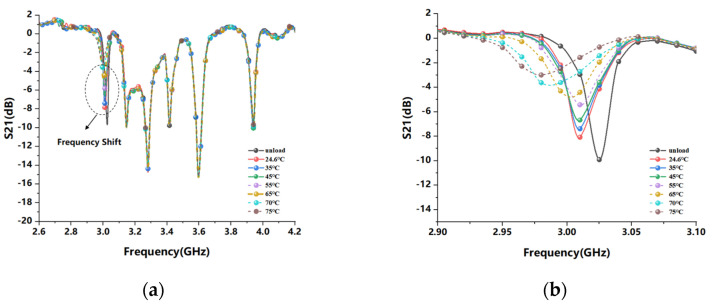
CSRR temperature sensing unit resonance response: (**a**) S21 resonance response curve change of tag sensor; (**b**) Partial amplification CSRR resonance unit S21 curve change.

**Figure 14 sensors-22-06027-f014:**
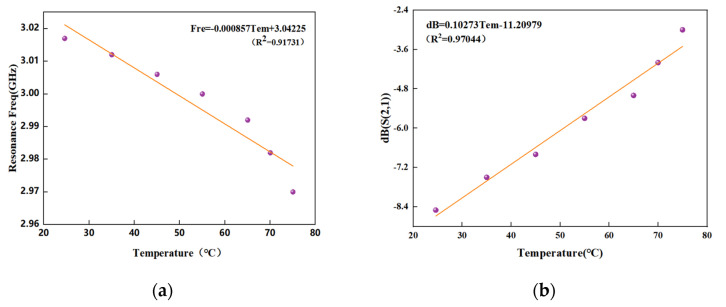
Linear normalized fitting of CSRR resonator temperature sensitivity: (**a**) CSRR resonant frequency changes with temperature; (**b**) CSRR resonant frequency peak depth changes with temperature.

**Figure 15 sensors-22-06027-f015:**
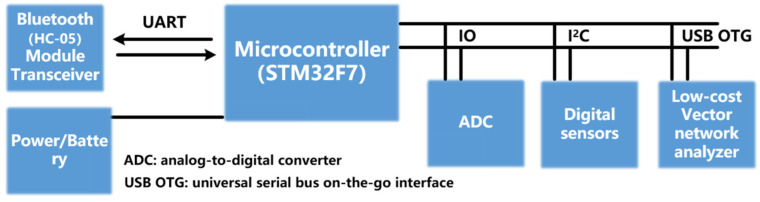
Dynamic spectrum detection and data transmission system architecture.

**Figure 16 sensors-22-06027-f016:**
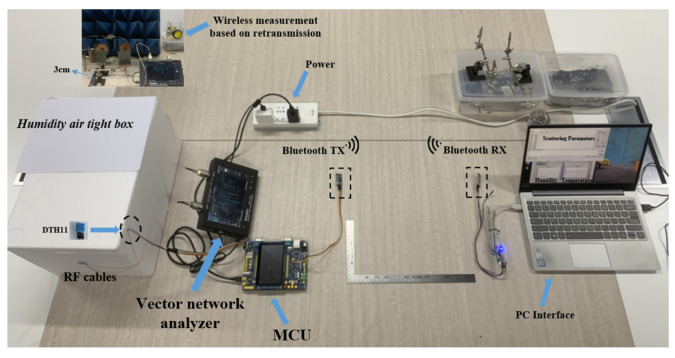
RFID tag dynamic spectrum monitoring and data wireless transmission experiment.

**Table 1 sensors-22-06027-t001:** Chipless RFID tag size design (unit: mm).

**Parameter**	*L*1	*L*2	*L*3	*W*1	*W*2	*W*3
**Value**	9.6	7.5938	10	7.2	7.5938	7.5
**Parameter**	*d*1	*d*2	*d*3	*C*1	*C*3	*Wc*1
**Value**	0.6	0.255	0.625	1.2	1.25	2.826
**Parameter**	*Lc*1	*dg*1	*dg*2	*Wf*	*Width*	*Length*
**Value**	6	0.1	0.1	2.4	40	50

**Table 2 sensors-22-06027-t002:** The 6-bit chipless RFID tag design size value (unit: mm).

**Parameter**	*L*4	*L*5	*L*6	*L*7	*L*8	*W*4	*W*5	*W*6	*W*7
**Value**	8.019	10.08	10.72	11.04	11.68	8.019	7.56	8.04	8.28
**Parameter**	*W*8	*C*5	*C*6	*C*7	*C*8	*d*4	*d*5	*d*6	*d*7
**Value**	8.76	1.26	1.34	1.38	1.46	0.27	0.63	0.67	0.69
**Parameter**	*d*8	*Lc*2	*Wc*2	*dg*3	*h*	*Wf*	*Length*		*Width*
**Value**	0.73	6.336	2.98	0.2	0.79	2.4	50		50

**Table 3 sensors-22-06027-t003:** Comparison of the proposed tag with other chipless multi-parameter sensors in the literature.

Resonator Type	Size (mm^2^)	Smart Materials	Sensing Parameter	Range	Sensitivity	Reference
Split box resonator	14.38 × 156	None	Crack and moisture	NA	NA	[1]
ELC and U-shaped slots resonators	6.8 × 15	PVA and Kapton	Humidity and encoding	35–85%RH, 6-bit	NA	[8]
Etched circular slots resonator	7.4 × 7.4 × π	MWCNT and Kapton^®^ HN substrate	Humidity and gas	40–70%RH	NA	[11]
ELC and U-shaped slots resonator	8 × 25	PVA and Phenanthrene	Humidity threshold temperature and encoding	35–85% RH, 65–95 °C, 3-bit	NA	[16]
Complementary split ring resonator (CSRR)	8 × 9	GO@PI and HTCC substrate	Humidity, temperature and pressure	20–90% RH, 25–300 °C, 10–300 kPa	389 kHz/% RH, 1.52 MHz/% RH(60–90%RH), 133 kHz/°C, 107.78 kHz/kPa	[19]
Asymmetric circular four split ring resonator (ACiSRR)	25 × 25	GO, rGO and Chitosan	Humidity, temperature and pH	NA	NA	[24]
Split ring resonator (SRR)	18.5 × 46	SWCNT	Temperature and gas	30–60 °C, 500–20,000 ppm	36.9% (RCS) for 30 °C,12.2% (RCS) for 20,000 ppm	[31]
U-shaped resonator and L-shaped resonators	15 × 35	Rogers 6010.2LM substrate	Crack and temperature	0.1–0.5 mm, 25–65 °C	NA	[32]
Multistate-coupled line resonators	NA	Stanyl and Kapton	Humidity and temperature	35–85% RH	NA	[33]
Split ring resonator (SRR)	NA	Ag@MoS_2_ and polyimide (PI) substrate	Humidity and gas	0–60% RH, 0–100 ppm	0.097% ppm^−1^	[34]
CSRR and ELC resonators	50 × 50	PVA and PVA-rGO	Humidity, temperature and encoding	30–85% RH, 24.6–75 °C, 4-bit	2.18 MHz/RH, 898.63 KHz/°C, 0.10273 dB/°C	This work

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
