# Peer review of "Characteristic Analysis of a Chipless RFID Sensor Based on Multi-Parameter Sensing and an Intelligent Detection Method"

_sensors, 2022, doi:10.3390/s22166027_

Round 1

Reviewer 1 Report

The paper presents a chipless sensor model that allows simultaneously reading the values of two environmental parameters and transmitting a 4-bit code for sensor identification. Resonant operation of each section is provided with layout elements specific for this type of applications: complementary split ring resonators (CSRRs) and electric field LC-coupled resonators (ELCs) compatible with microstrip transmission lines. In Section 3.3 a system for dynamic, mobile data acquisition and transmission is also described. Unfortunately, the paper does not present much detail on: (i) permissible distances between the interrogation equipment and the sensor (this should be equipped with compatible antennas), and (ii) how the system allows selection between two sensors that are programmed with different codes but are at equal distances from the interrogation equipment (detailed explanation).All this information would have made the work much more interesting and appreciated.

Other comments and questions:

[row 97] By "the resonant units are not easily coupled" did the authors mean that the resonators are weakly coupled?

[rows 208-210] The authors consider that “CSRR resonators are prone to generate multiple resonant frequencies in the low frequency region because of their higher Q value”; please explain this statement.

[rows 243-244] The authors claim that "the corresponding CSRR resonator is removed"; what is the removal technique for CSRRs?

[rows 269-270] The authors say: “this paper made PVA colloquial film and pasted it on the resonance unit”. (1) It’s not the paper that made the film. (2) The word "colloquial" is not the most appropriate in this (technical) context. (3) Could “pasted” be replaced with “transferred” (see also row 282)?

[rows 329-332] Related to humidity sensor’s reversibility: what are the actual response and recovery times for the PVA-based moisture sensor?

[row 352] The authors say: “this paper places an intrusive heater”. It’s not the paper….

[rows 380-403] Correlate the correspondence between text and figure numbers.

[rows 394-397] Use “frequency decrease” (or similar) instead of “frequency shift to the left”, “frequency shifts leftward”, “leftward frequency shift”.

English language. The formulation in the following fragments of text should be revised: rows 233-235; 292-293; 320-321; 413-424.

Other editing issues: (i) full name of each term to which an acronym is associated must appear with the first mention of that acronym, both in the Abstract (see row 22 for VNA) and in the text of the paper; (ii) use space between text and any bracket type.

Row number

Actual text

Suggestion(s)

100

50 microstrip transmission

50Ω microstrip transmission

124

n is the n modes

n is the nth resonant (or resonance) mode

142

main parameter codes

main parameter dimensions

266

Certain

certain

302

Figure 10

Figure 11

313

sag depth

penetration depth

338

smearing

spread (or spreading)

373

Among them, ε is

where ε is

388

residence time

settling (or response) time

402

measured area

measured range

415

bluetooth

Bluetooth (B must be capital letter)

Author Response

We highly appreciate you for the excellent comments and suggestions, which are valuable and helpful for revising and improving our manuscript. Our responses are presented as follows and all changes to the manuscript have been marked in yellow.

Reviewer 2 Report

This paper proposed chipless radio frequency identification (RFID) with temperature and humidity sensors. Several experiments were conducted to evaluate the performance of the proposed method. The result seems OK, but the following comments can be addressed before it's publication:

-A summary of contributions should be presented at the end of the introduction section.

-Literature studies section should be added. Please summarize the related studies regarding RFID sensors, applications of RFID, etc., and contrast their studies with yours. A table would be enough and at least 10 recent literature studies could be added.

-Please provide the optimum frequency with regard to getting the accurate temperature and humidity from the RFID sensors.

-Cost analysis could be added to know the cost of developing the sensors and devices. Please include the link to the sensors/devices whenever possible.

-Finally, the limitations and future studies should be presented in the conclusion section.

Author Response

(The authors gave the same response as above.)

Reviewer 3 Report

A brief summary:

The authors proposed multi-parameter a chipless RFID sensor based on microstrip coupling, which supports 4-bit ID code and integrates two detection functions of temperature and humidity.

Strengths:

-         The article is written in an appropriate way.

-         The paper is well-written, and presents comprehensive analysis results.

-         The conclusions are justified and interesting for the readership of the journal.

Weaknesses.

-          Motivation and novality should be mentioned in the introduction.

-          Contribution of the work should be mentioned in the introduction.

Recommendations:

-         The language is good. This does not preclude reviewing the paper again, linguistically.

-         The authors are suggested to try to better show the contribution of the paper in the introduction section, as well as to give more elaborate discussion.

-         Please, revise the “Abbreviations” section to ensure that all abbreviations are used.

Decision:

-         Major revision.

Author Response

(The authors gave the same response as above.)

Round 2

Reviewer 1 Report

The paper may be published in its current form.

Reviewer 2 Report

The author has addressed all previous comments.

Reviewer 3 Report

Required modification have been done.